# Image-Based Multi-Agent Reinforcement Learning for Demand–Capacity Balancing

**Sergi Mas-Pujol *** , **Esther Salamí** and **Enric Pastor**

Department of Computer Architecture, Escola d'Enginyeria de Telecomunicació i Aeroespacial de Castelldefels (EETAC), Universitat Politècnica de Catalunya (UPC), Esteve Terradas 7, 08860 Castelldefels, Barcelona, Spain
* Correspondence: sergi.mas.pujol@upc.edu

**Abstract:** Air traffic flow management (ATFM) is of crucial importance to the European Air Traffic Control System due to two factors: first, the impact of ATFM, including safety implications on ATC operations; second, the possible consequences of ATFM measures on both airports and airlines operations. Thus, the central flow management unit continually seeks to improve traffic flow management to reduce delays and congestion. In this work, we investigated the use of reinforcement learning (RL) methods to compute policies to solve demand–capacity imbalances (a.k.a. congestion) during the pre-tactical phase. To address cases where the expected demands exceed the airspace sector capacity, we considered agents representing flights who have to decide on ground delays jointly. To overcome scalability issues, we propose using raw pixel images as input, which can represent an arbitrary number of agents without changing the system's architecture. This article compares deep Q-learning and deep deterministic policy gradient algorithms with different configurations. Experimental results, using real-world data for training and validation, confirm the effectiveness of our approach to resolving demand–capacity balancing problems, showing the robustness of the RL approach presented in this article.

**Keywords:** air traffic flow management; demand–capacity balancing; reinforcement learning; multi-agent; deep Q-learning; deep deterministic policy gradient; convolutional neural networks



## 1. Introduction

Congestion problems arise in situations where limited resources have to be shared simultaneously by multiple agents. They are present in a wide variety of domains in the modern world, and they have drawn much attention in Artificial Intelligence (AI) research [1,2]. Air Traffic Management (ATM) is one domain where congestion problems appear naturally, introducing extra costs and uncertainty to operations scheduling. Concretely, congestion problems appear when the expected number of aircraft (airspace demand) exceeds the maximum number of flights that the Air Traffic Controllers (ATCOs) can safely manage for a particular airspace sector (capacity). This is known as the Demand–Capacity Balancing (DCB) problem or process [3].

Initially, demand–capacity imbalances are solved via airspace management or flow management solutions. However, when no solution is enough, Air Traffic Flow Management (ATFM) regulations are implemented issuing extra ground delays to the necessary flights. This cascade of events increases the uncertainty regarding the scheduling of operations, costs [4], and unforeseen effects on the entire system. Furthermore, these events present further negative effects for the ATM stakeholders, including environmental effects, customer satisfaction, and loss of reliability.

In the European Air Traffic Control (ATC) network, ATFM delays are imposed by the Computer Assisted Slot Allocation (CASA) algorithm [3,5], which is a heuristic algorithm based on the principle of first-planned-first-served. In 2018, prior to COVID-19, at the European Civil Aviation Conference (ECAC) level, the number of flights increased by +14.6%,

corresponding to 1.4 million additional flights in 2018 compared to 2013. At the same time, en-route ATFM delays more than doubled compared to 2017 (+104%). As a result, 9.6% of the flights issued ATFM delays with an increment of 1.74 min per flight [6].

Nowadays, demand–capacity imbalances are difficult to predict during the pre-tactical phase (from several days to a few hours prior to operations) because of the uncertainty in the operational information. Indeed, DCB is a two-stage problem: first, it is necessary the identification of the overloaded regions (demand greater than capacity); second, the CASA algorithm assigns new departure slots smoothing the demand to meet the pre-defined capacity.

Previous research on ATFM regulations focused on the detection and/or resolution of DCB issues, optimization algorithms, development of new performance metrics, or novel techniques. For instance, in previous research [7,8], we proved the potential of supervised Machine Learning (ML) models to predict which sectors were going to be regulated, Reference [9] used ML models to predict the evolution of the ATFM delay for regulated flights, and Reference [10] investigated the detection of regulations due to convective weather and the associated airspace performance characteristics.

Other works investigated optimization techniques to find optimal resource utilization. Reference [11] presented an optimization algorithm to minimize the propagation of ATFM delays to subsequent flights, Reference [12] introduced an integer programming model for strategic redistribution of flights to respect nominal sector capacities in short computation times for large scales and Reference [13] investigated a new technique that could improve airspace capacity usage and reduce ATFM delays by improving the slot allocation process of CASA to avoid wasted capacity (empty slots) in regulated sectors.

On the other hand, several works attempted to study the downstream effects of ATFM regulations and propose resolution techniques. Reference [14] used gradient-boosted decision trees to predict the likelihood of a regulated flight rerouting to mitigate the ATFM delay, and Reference [15] proposed using speed reduction on air to absorb ATFM delay at no extra cost. Most recent works on the resolution of DCB issues focused on the use of Reinforcement Learning (RL) techniques. For instance, Reference [16] proved it was possible to both identify and resolve DCB problems by comparing three RL algorithms for the pre-tactical phase. Similarly, Reference [17] is the result of a set of publications where the DCB was formulated as a hierarchical Multi-Agent Reinforcement Learning (MARL) decision problem with different levels of abstraction. However, one important drawback of this MARLs approach is that a different agent controls each flight, presenting a severe scalability problem, as hundreds or even thousands of different agents would be required to handle the full European Air Traffic Management Network (EATMN).

In response to the previous scalability limitations, Reference [18] presented a collaborative Multi-Agent Asynchronous Advantage Actor-Critic (MAA3C) framework with embedded supervised and unsupervised Neural Network (NN), where only flights crossing airspace sectors with already identified demand–capacity issues were regarded as the candidate agents. This approach improved the scalability and generalization of the system, being able to handle a varying number of agents. As an extension of the scalability issues, Reference [19] reviewed different deep MARL methods, examining their ability to scale up to large agent populations (from hundreds to several thousands of agents). The main conclusion drawn with respect to possible scalability issues is the importance of parameter sharing in large agent populations. It is impractical to train thousands of independent networks for each agent or to utilize an approach whose input size would explode as the number of agents and their observations grow larger.

Similar research has been conducted outside the EATMN. In the USA network, Reference [20] developed a MARL system for ATM integrated with an air traffic flow simulator—FACET. In Reference [21], the authors presented a distributed decision support system for tactical ATFM in Brazil and traffic flow managers experts analyzed the solutions proposed by the system.

Traffic growth and changes in traffic patterns have caused increasing congestion and delay in European airspace. The Central Flow Management Unit (CFMU) continually seeks and develops methods to improve traffic flow management to reduce delays and congestion [5]. To this end, and taking into account the available literature, as a research question, this article investigates whether a RL system is able to solve demand–capacity imbalances for specific airspace sectors without sharing explicit information between agents, using an approach whose size does not depend on the number of agents and with a fixed size of the observation states.

This article formalizes the ATFM problem as a collaborative MARL system where homogeneous agents, representing flights, aim to decide on their ground delay jointly with the other flights, while not having direct information about the preferences of other flights. The goal is to smooth already-identified pre-tactical DCB problems in a specific airspace sector using images as input to the system and ensuring efficient utilization of the airspace. The usage of images allows the system to extract its own features for the problem instead of manually deciding which ones are more representative, the input size is independent of the number of agents, and it provides a fixed size of the states ensuring good scalability. Moreover, the images allow the agents to have indirect information about the other flights.

To smooth the demand–capacity imbalances, we investigated two types of RL algorithms: first, algorithms based on discrete actions; second, algorithms based on continuous actions. In both cases, we used a homogeneous population of agents to ensure their behavior was the same. Furthermore, the agent-based paradigm introduced in this article tried to emulate the first-planned-first-served basis used in the current ATFM approach. Only flights outside the regulated sector were candidates to be agents, ensuring that only flights outside the airspace sector would be delayed.

Experimental results demonstrated the effectiveness and robustness of the novel approach presented in this paper. The work presented in this article aimed to be the first step toward devising multi-agent methods for deciding on delay policies using images. The contributions made in the paper are as follows:

- The DCB problem was formulated as a MARL based on the use of images to improve scalability.
- Two different types of MARL algorithms were studied, one using continuous actions (DQN) and another one using discrete actions (DDPG).
- For the DDPG algorithm, three different approaches for exploration noise (random values used for trial-and-error search) were analyzed.
- All configurations were trained and evaluated using real-world data.

This paper is organized as follows: Section 2 provides relevant background about RL, focusing on Q-learning and deterministic policy gradient. Section 3 presents the problem formulation. Section 4 describes the experimental setup focusing on the parameters of the algorithms. Section 5 exhibits the experimental results obtained and their analysis. Section 6 summarizes the work presented in this article and addresses future work.

## 2. Reinforcement Learning

RL problems consist of learning what to do (how to map situations to actions) to maximize a numerical reward signal. The agent is not told which actions to take, but it must discover which actions yield the most reward by trying them. Notice that actions may affect not only the immediate reward but also the following states and, through that, all subsequent rewards. These two characteristics, trial-and-error search, and delayed reward, are the two most important distinguishing features of RL. Therefore, a learning agent must be able to sense the state of the environment, take actions that affect the state, and have a clear goal (or goals) relating to the state of the environment [22]. This interaction is depicted in Figure 1.

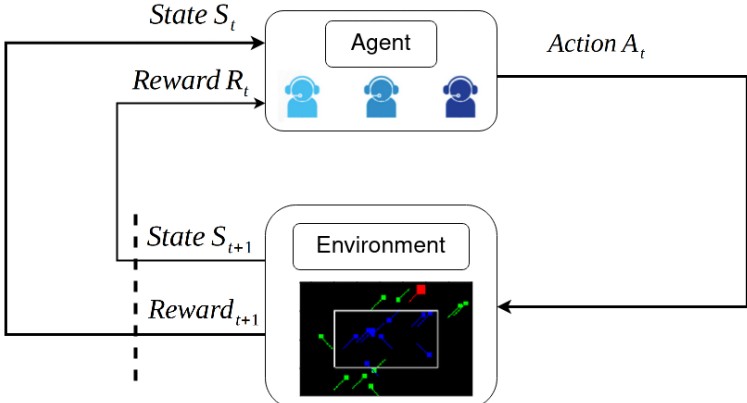

**Figure 1.** Interaction between the different elements in a RL system (adapted from [23]).

One of the challenges that arise in RL is the trade-off between exploration and exploitation. To obtain as much of a reward as possible, a RL agent must prefer actions that it tried in the past and found to be effective. However, to discover them, it has to try actions it has not selected previously. The agent has to exploit what it has already experienced to maximize the reward, but it has to explore to make better action selections in the future.

Beyond the environment and the agent, we can identify four main sub-elements:

- Policy: roughly speaking, it is a mapping from the states of the environment to actions.
- Reward signal: it defines the goal of the RL problem. It defines what is "good" in an immediate sense.
- Value function: specifies what is "good" in the long run. Roughly speaking, it is the total reward an agent can expect to accumulate over the future, starting from a particular state.
- Model: it mimics the behavior of the environment. It allows inference to be made about how the environment will behave.

### 2.1. Single-Agent Reinforcement Learning

A RL problem for a single agent interacting with an environment can be formalized as a finite Markov Decision Process (MDP) described by the tuple $(\mathcal{S}, \mathcal{A}, P, R)$, where $\mathcal{S}$ is the set of states of the environment, $\mathcal{A}$ is the set of actions the agent can take, $P$ is the transition function, being $P(s'|s, a)$ the probability of transitioning to $s' \in \mathcal{S}$, by applying $a \in \mathcal{A}$ in $s \in \mathcal{S}$, and $R$ is the reward function. Notice, in a finite MDP, the sets of states, actions, and rewards ($\mathcal{S}$, $\mathcal{A}$, and $R$) have a finite number of elements.

At each time step, the reward is a simple number, $R_t \in R$. However, the agent's goal is to maximize its cumulative reward $G$. That is, maximize both the immediate reward and cumulative reward in the long run. Thus, the rewards we set up must truly indicate what we want to accomplish. The cumulative reward, also referred to as return, can be defined as follows:

$$G = \sum_{t=0}^{\infty} \gamma^t R_t \tag{1}$$

where $\gamma$ is a parameter, $0 < \gamma < 1$, called the discount rate. It determines how much the agent cares about immediate rewards relative to distant ones.

The RL system aims to find the optimal policy $\pi_*$, which maximizes the expected commutative reward. Let us define the value function of a state $v_\pi(s)$, for a policy $\pi$ (which may not be optimal), as the expected return when starting in state $s$ following policy $\pi$. For MDPs, we can define $v_\pi(s)$, formally by:

$$v_\pi(s) = \mathbb{E}[G_t|S_t = s] \tag{2}$$

Similarly, we define the value of taking action $a$ in state $s$ under a policy $\pi$ as $q_\pi(s, a)$, providing the expected return:

$$q_\pi(s) = \mathbb{E}[G_t | S_t = s, A_t = a] \tag{3}$$

There is always at least one policy that is better than or equal to all other policies. The optimal policy. Although there may be more than one, all the optimal policies are denoted by $\pi_*$. They share the same state-value function, called the optimal state-value function, denoted $v_*$ and defined as:

$$v_*(s) = \max_\pi v_\pi(s) \tag{4}$$

for all $s \in \mathcal{S}$.

Optimal policies also share the same optimal action-value function, denoted $q_*$ and defined as

$$q_*(s) = \max_\pi q_\pi(s, a) \tag{5}$$

for all $s \in \mathcal{S}$ and $a \in \mathcal{A}$.

Therefore, the optimal policy $\pi_*$ selects what action maximizes the expected commutative reward. If the optimal action-value function $q_*(s, a)$ is known, the best action in the state $s$ is given by:

$$\pi_*(s) = arg \max_{a \in \mathcal{A}} q_*(s, a) \tag{6}$$

The two main approaches used to obtain the optimal policy are policy iteration, which manipulates the policy directly, and value iteration, which aims to find an optimal value function adopting a greedy policy.

### 2.2. Multi-Agent Reinforcement Learning

A MARL system involves a set of $\mathcal{N}$ interacting agents, which can be cooperative, competitive, or both. It can be described by the tuple $(\mathcal{N}, \mathcal{S}, \{A_i\}_{i \in \mathcal{N}}, \{O_i\}_{i \in \mathcal{N}}, P, \{R_i\}_{i \in \mathcal{N}})$.

At every time step, each agent $i \in \mathcal{N}$ observes a partial representation of the environment $o_i \in \mathcal{O}_i$, and performs an action $a_i \in \mathcal{A}_i$ determined by a policy function $\pi_i$. Then, when an action is taken, the environment evolves to a new state $s' \in \mathcal{S}$, according to the transition function $P$. This transition function depends on the current state and the joint action of all agents. Finally, the reward that each agent receives is given by the reward function. For instance, agents typically share the reward in a cooperative RL.

One possible approach for MARL is to train independent agents. However, this simple approach does not perform well in practice [24]. To overcome these limitations, in [25,26], each agent has its centralized critic, only used during learning, which approximates and learns the action-value function given the observations and actions of all agents. However, the critics require the actions and observations of all agents as input. Consequently, their complexity is proportional to the number of agents.

A different solution is proposed in [27] to mitigate this scalability issue. In this case, the agents learn an individual action-value function based on their local observations, and the sum of these functions approximates the centralized joint action-value function.

### 2.3. Q-Learning

Q-learning [23] is one of the most well-known algorithms based on value iterations. It makes use of a Q-table, which typically has the shape [states, actions], and each Q-value $Q(s, a)$ represent the quality of taking as action $a \in \mathcal{A}$, in $s \in \mathcal{S}$. Thus, Q-learning was designed to work with discrete actions.

At each time step $\Delta t$, the agent observes the current state $s_t$ and chooses the action $a_t$ with the highest Q-value in that state. After applying the selected action, the agent receives a reward $r_t$, enters on new state $s_{t+1}$, and the Q-value is updated using Equation (7):

$$Q(s_t, a_t) \leftarrow Q(s_t, a_t) + \alpha \left( r_t + \gamma \max_a Q(s_{t+1}, a_t) - Q(s_t, a_t) \right) \tag{7}$$

where $r_t$ is the reward received when moving from state $s_t$ to $s_{t+1}$, $\alpha \in [0, 1]$ is the learning rate and $\gamma \in [0, 1]$ is the discount factor.

According to Equation (7), the agent adopts a greedy strategy by constantly selecting the actions with the largest Q-value. In that case, it exists the risk of adopting a sub-optimal solution by converging to a local minimum. The $\epsilon$-greedy strategy is widely used to properly explore the state space, where $\epsilon$ corresponds to the probability of choosing a random action. Typically, $\epsilon$ is usually initialized to 1 to force high exploration at the beginning, with a decay rate over time to ensure exploitation at the end of the training.

One limitation of this well-known algorithm is the rapid growth of dimensionality in the state space. The traditional solution is deep Q-learning [28], which uses a NN to approximate the Q-values. However, instead of training the NN with the sequence of experiences as they occur during the simulations, they are saved in what is usually called the experience replay buffer. Using a buffer prevents the agent from forgetting past experiences as time evolves and breaks the correlation between consecutive experiences. Finally, a target network is used to stabilize the learning. The target network is the result of periodically replacing its weights with the ones from the online network used to select the action greedily.

### 2.4. Deterministic Policy Gradient

Deterministic Policy Gradient (DPG) [29] is an actor–critic RL algorithm, used for continuous actions, which learns a deterministic policy function and a value function simultaneously, from an exploratory behavior.

It is not possible to straightforwardly apply Q-learning to continuous action spaces because finding the greedy policy would require optimization of $a_t$ at every time step, which is too slow to be practical with large, unconstrained functions approximators, and nontrivial action spaces [29]. The DPG algorithm uses an actor as the current policy to map states to a specific action. The critic determines the expected reward for an agent starting at a given state and acting according to the previous policy.

As with Q-learning, to learn and generalize on large-scale state spaces, it is required to introduce non-linear function approximators, which means that convergence is no longer guaranteed. However, such approximators appear essential in those scenarios [30] that presented a modification to DPG from [31], inspired by the success of Deep Q-Learning (DQN), allowing the use of NN function approximators. This implementation is called Deep Deterministic Policy Gradient (DDPG), and it was proved that the algorithm could learn policies "end-to-end" directly from raw pixel inputs. Target networks are used to add stability to the training, and an experience replay buffer is used to learn from experiences accumulated during the training.

### 3. Problem Formulation

In the current European ATC network, ATFM delays are particularly complex problems. When flights are affected by an ATFM delay, they are issued with a Calculated Take-Off Time (CTOT), which indicates the new time windows for the flight to depart (from 5 min before the CTOT to 10 min after). This new CTOT is computed by the CASA algorithm, and if the flights cannot depart within this window, the ATFM slot will be missed and a new one will be assigned.

ATFM regulations are located at specific airspace traffic volumes (which can be informally defined as a portion of airspace linked to a sector) where a demand–capacity imbalance is detected. Nowadays, the methodology used to identify where ATFM regulations are required is purely human and does not rely on automation. Air Navigation Service Providers (ANSPs) define two capacities for the sectors, which have to be interpreted by the Flow Manager Position (FMP): the sustained capacity and the peak capacity. The sustained capacity indicates the maximum number of flights that can be operated for a particular

time window, while the peak capacity indicates the maximum value for a specific instant of time. Close to the day of operation, capacities are defined based on the Occupancy Count (OC), which considers the expected number of flights inside the traffic volume.

It is possible to have multiple demand–capacity imbalances in the network simultaneously. However, the general principle is that a flight subject to several ATFM regulations is given the delay of the most penalizing regulation.

### 3.1. Assumptions

In this work, the following assumptions are considered to define the ATFM delay (a.k.a. ground delay) system for specific traffic volumes:

1. The airspace sectors with a demand–capacity imbalance are known (interval of time with overload, location, and capacity), and squares can be used to approximate their shape.
2. Pre-tactical flight plans are available for each flight before any regulation is applied. The flight plans contain the Scheduled Off-Block Time (SOBT) and the route of the flight. Additionally, it is assumed constant speed for each of the segments composing the routes.
3. There is one type of agent. There are no aircraft with priority.
4. Financial costs imposed on commercial entities resulting from ATFM decisions are negligible.

There is a deviation from traditional state-of-the-art problems by assuming the demand–capacity imbalances are already known for the sector of study. Assumption 1 was used because this work focused purely on the resolution of DCB issues. Only historical data from regulated intervals and sectors were used. Moreover, related to assumption 1, the approximation of the sector's shape as squares aimed to reduce the implementation complexity in this preliminary study.

Related to assumption 2, we considered the constant speed per segment defining the routes because they only contained information about the starting/ending location and time. By assuming a constant speed between the origin and end of the segments, it is possible to interpolate the location of the flights at intermediate timestamps (see [32,33] as other examples of interpolation).

Assumption 3 aspires to create a prototype that is as fair as possible for all the operators. A homogeneous population of agents guarantees that all flights are treated equally. However, using heterogeneous populations of agents in future work could be interesting from an optimization point of view. For instance, the use of different populations between domestic or international flights, or to prioritize transit flights to avoid possible downstream effects such as missing connections. Similarly, assumption 4 is used to emphasize that this prototype focuses on the current used Key Performance Indicators (KPIs), although they could be extended according to additional requirements if needed.

### 3.2. Action Variable

The action variable in this problem corresponds to selecting the ground delay that an aircraft will receive due to a demand–capacity imbalance. At each step $\Delta t$, each agent $i \in \mathcal{N}$ has an associated action variable $a_t^i \in \mathcal{A}$, where $a_t^i$ is the ATFM ground delay.

For discrete action algorithms, the action variable can be defined as:

$$a_t^i \in \mathcal{A}, \quad \mathcal{A} \in \{0, 5, 10, 15\} \tag{8}$$

While for continuous action algorithms, the action variable can be defined as:

$$a_t^i \in \mathcal{A}, \quad \mathcal{A} \in [0, 15] \tag{9}$$

### 3.3. State Variable

The state vector $s_t^i \in \mathcal{S}$ includes the information that the population of agents $\mathcal{N}$ uses to determine the actions. Each state $s_t^i$ is defined per flight candidate to be an agent and step of the system.

One of the primary challenges associated with MARL is problem representation. The challenge is in defining the problem in such a way that an arbitrary number of agents can be represented without changing the architecture of the DQN or DDPG. To solve this problem, we propose the usage of image-like tensors where each channel in the images encodes a different set of information from the global state. This representation allows us to take advantage of Convolutional Neural Networks (CNNs), which have been shown to work well for image classification tasks [34] and competitive MARL systems based on images [35].

The image tensor is of size HxWx3 (shown in Figure 2), where H is the height, W is the width of our two-dimensional images, and three is the number of channels in the image. The channels can be broken down in the following way:

- Inside channel: Contains the representation of the flights inside the sector being regulated.
- Outside channel: Contains information about the flights outside the sector of study, i.e., the flights that may be delayed.
- Self channel: Contains information about the agent making the decision.

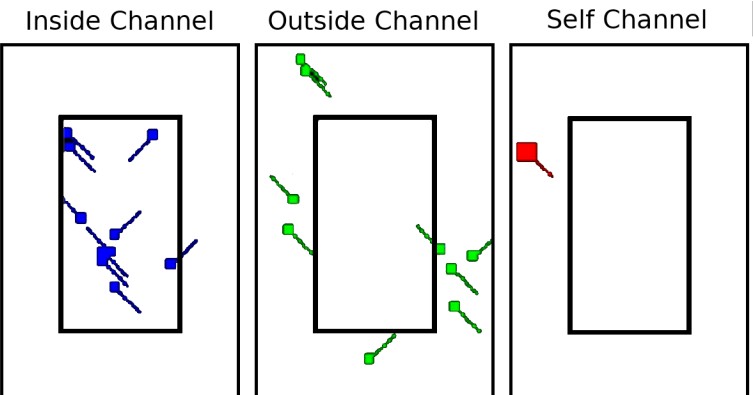

**Figure 2.** Three channels, image-like, represent the input states of the RL system. The inside and outside channels encode information about the flights. The self-channel encodes information about the agent that is taking action. The inner square represents the shape of the sector.

Note that the three channels are depicted with white backgrounds for clearness, but they encode zero pixel values. The non-zero pixel values encode the locations of the flights, their headings, and the approximate shape of the sector.

### 3.4. State Transition

The state transition defines a set of conditions that determine how the state $s_t^i \in \mathcal{S}$ evolves along the steps. With every step, the aircraft candidates (to be agents) decide whether they are going to issue ATFM delays. Three conditions must be verified to ensure a proper transition between states.

The first condition that must be verified is related to the regulations used for the training. Each episode will start using information from a randomly selected historical regulation, and the environment will evolve for a time period $TP$ equal to 60 min with a timestep $\Delta t$ equal to one minute. Thus, from the randomly selected regulation, we must guarantee that the regulations will be active for more time than the $TP$.

The second condition to consider is related to the delay. For each state variable $s_t^i \in \mathcal{S}$, the agent $i$ will produce a new action to cooperatively decide its own ground delay to ensure that the demand meets the capacity. Actions equal to zero imply no delay for the

flight moving forward on the predefined trajectory. However, if the delay differs from zero, the new delay is added to possible previous delays (cumulative delay).

The last required consideration is related to how the flight is assumed to move forward. A trajectory $T \in \mathcal{T}$ is a time series of segments of the form:

$$T = \{(ID_l, begin_{t_l}, end_{t_l}, lat\_begin_{t_l}, lon\_begin_{t_l}, lat\_end_{t_l}, lon\_end_{t_l})\} \quad l \in [1, m] \quad (10)$$

where $ID_l$ is the identifier of the segment, $begin_{t_l}$ the initial timestamp of the segment, $end_{t_l}$ the end timestamp of the segment, $lat\_begin_{t_l}$, $lon\_begin_{t_l}$ the initial latitude and longitude of the segment, $lat\_end_{t_l}$, $lon\_end_{t_l}$ the end latitude and longitude of the segment, and $l$ is the number of segments used to define the trajectory.

For each of the segments, we assumed constant speed. Therefore, the expected velocity of the flight in a particular segment can be defined as follows:

$$v_{ID_l} = \frac{f(lat\_end_{t_l}, lat\_begin_{t_l}, lon\_end_{t_l}, lon\_begin_{t_l}}{end_{t_l} - begin_{t_l}} \quad (11)$$

where $f$ is a function that computes the distance between two pairs of coordinates.

Finally, we can compute the aircraft's location at any timestamp, knowing the required segment to use, assuming constant speed in the segments, and taking into account the imposed ATFM delay.

### 3.5. Problem Constrain

A set of operational constraints is associated with the sectors' capacity, whose violation results in DCB problems (congestion). This violation occurs when $OC_{TP} \geq C$, where $OC_{TP}$ is the demand of a predefined counting period and $C$ is the capacity of the sector. Although this violation will result in a DCB issue, it must be taken into account that two capacities are defined: sustain and peak capacity. The sustained capacity can be exceeded for a short period, while the peak capacity should not be exceeded. Furthermore, for each new episode, it must be guaranteed that there is no demand–capacity imbalance for the selected random starting timestamp of the ATFM regulation. Ensuring that the agents are not directly penalized due to the demand–capacity imbalance without being able to perform any action.

Another constraint is related to which flights are considered candidates to be agents. Thus, the flights can be delayed in each step. To emulate as close as possible the current approach based on first-in-first-served used by CASA, for each step, only those flights outside the congested sector are considered agents.

The last constraint of the system is related to the ATFM delay itself. The maximum delay an agent can impose on a flight per step is equal to 15 min. This aims to allow agents to delay flights without imposing huge penalization.

### 3.6. Objective Function

Demand reduction is one of the main goals in DCB during the pre-tactical phase. The objective is to try to smooth the traffic and meet the expected demand with the predefined capacity of the airspace sector. The objective function can be defined with Equation (12), which corresponds to minimizing the ATFM delay while trying to ensure that the demand meets the sector's capacity for the counting period.

$$\min_{t \in TP} \mathbb{E} \left\{ \sum_{i=0}^{\mathcal{N}} D_i \left( s_t, \pi^*(s_t) \right) \right\} \cup V_t \leq C \quad (12)$$

where $D_i$ is the ATFM delay of agent $i$, $\mathcal{N}$ is the population of the agent, $s_t$ is the state of the system at step $t$, $\pi^*$ is the optimal ATFM delay policy, $V_t$ is the OC of the sector, and $C$ is the capacity of the sector.

## 4. Experimental Setup

This section details the developed DQN and DDPG algorithms, focusing on the dataset used to train the agents, the RL elements, and the parameters of the algorithms.

### 4.1. Dataset

The proposed RL algorithms are trained using pre-tactical information from Aeronautical Information Regulation and Controls (AIRACs). Concretely, we used data from June, July, August, and September 2019.

In the EATMN, a wide variety of regulations are applied due to many reasons across different traffic volumes. The study done in this article focuses on *C-ATC Capacity ATFM regulations*, which are those regulations purely related to demand–capacity imbalances. Other types of regulations can be related to convective weather or military operations. Moreover, because of the huge number of sectors, we focused our attention on the Maastricht Upper Area Control Centre (MUAC) region. In particular, to the sector *EDYYBOLN* with the associated traffic volume *MASBOLN*. The main reason behind the selection of this particular sector is because it is one of the most regulated airspace regions in the MUAC area, which will guarantee enough variety of samples to train the RL agents. The available dataset contains around 200 C-ATC Capacity ATFM regulations for en-route traffic along 71 different days, with a mean number of regulations per day equal to 1.7 and a mean duration per regulation of 97.08 min.

### 4.2. Reward Function

RL algorithms learn from the interactions with an environment, which provides a reward according to how good the agent's action was. The reward function is crucial because different reward structures will result in different system performances.

Previous research has investigated different reward functions. Typically, the literature shows that researchers mainly focused on delay and congestion without considering fairness impact on different commercial entities [20,36]. Similarly, [37] also took into account the amount of time the agents contributed to the demand–capacity imbalance. Fairness is usually measured by financial costs imposed on commercial entities resulting from ATFM decisions [38].

In our case, as a proof of concept using images, focus on delay and congestion. The reward function $G(z)$, written as Equation (13), consists of three main components: the number of flights delayed $C(z)$, the delay itself $D(z)$, and the demand–capacity ratio $I(z)$:

$$G(z) = -\beta M(z) - \delta D(z) - \lambda I(z) \tag{13}$$

where $z$ represents the system under evaluation, $M(z)$ and $D(z)$ represent delay, and $I(z)$ the congestion; $\beta$, $\delta$, and $\lambda$ are the weights used to adjust the income penalty in the evaluation function. Note that the reward function is based on penalties.

When ATFM delay is issued, the number of aircraft entering the airspace traffic volume is reduced; thus, the congestion is relieved. However, this restrictive measure has negative effects on the ATM network. Equation (14) counts the number of delayed flights and Equation (15) computes the total delays imposed:

$$M(z) = \Theta(\mathcal{N}) \tag{14}$$

where $\mathcal{N}$ is the population of agents, and $\Theta$ is a function that counts the number of flights that received ATFM delay.

$$D(z) = \sum_{t=0}^{P} d_{i \in \mathcal{N}}^{t} \tag{15}$$

where $D$ is the total ground delay, $\mathcal{P}$ is the counting period, and $r_i^t$ is the imposed ground delay at step $t$ for each agent $i \in \mathcal{N}$.

It is required to compute the number of aircraft at the current step to determine the congestion severity in the airspace sector; that is, the excessive number of aircraft in the sector. The congestion function $I(z)$ is given by:

$$I(z) = \begin{cases} (V-C)^{(V-C)} & V > C \\ 0 & Otherwise \end{cases} \tag{16}$$

where $V$ is the number of aircraft in the sector, and $C$ is the maximum number of aircraft in a sector that does not cause congestion (capacity). Note that the function is characterized exponentially with respect to the excessive number of aircraft in a sector.

### 4.3. Deep Q-Learning

In this work, the first RL algorithm we studied to optimize the ATFM delay was DQN. We followed the approach proposed in [39], which operates directly on RGB images to play Atari games, uses experience replay to store the agents' experiences, and uses a second target network.

At the beginning of each episode, a new initial state is set. Subsequently, for each step and flight candidate to be an agent, an action is chosen either randomly or greedily and stored in the replay buffer. In the first episode, the $\epsilon$-greedy strategy has an $\epsilon$ equal to 1 forcing agents to explore. However, this value linearly decreases until it reaches 0.01, ensuring the agents prioritize exploitation in the last episodes.

The input to the system is the images used to obtain the agent's experience tuple of the form $(s_t, a_t, r_t, s_{t+1})$, where $s_t$ is the starting image-like state, $a_t$ is the action taken, $r_t$ is the reward received, and $s_{t+1}$ is the new state of the system. The replay buffer stores the last 25,000 experience tuples, and batches with 64 samples are randomly selected to train the NN computing the target value and the respective loss. This loss is the minimum squared error of the predicted value and the target value, and the Adam optimizer [40] is used. After training the online network, the weights of the target network are also updated.

The input layer of the NN takes as input the $150 \times 100 \times 3$ images. The first layer convolves 32 ($8 \times 8$) filters with stride 4 and uses a Rectified Linear Unit (ReLU) activation function. The second layer is a batch normalization layer [41]. The third layer convolves 64 ($4 \times 4$) filters with stride 2 using a ReLU activation function. The fourth layer is a batch normalization layer. The fifth layer convolves 64 ($3 \times 3$) filters with stride 1 and uses a ReLU activation function. The sixth layer is a batch normalization layer. The final hidden layers are a fully-connected with 256 rectifier units and a droupout layer with a rate of 0.5. The output layer is a fully-connected linear layer with a single output for each valid action. The output of the NN corresponds to the predicted Q-values of the individual action for the input state. The main advantage of this type of architecture is the ability to compute Q-values for all possible actions in a given state with only a single forward pass through the network. Table 1 shows the remaining hyperparameters.

**Table 1.** Hyperparameters for the deep Q-learning algorithm.

| Hyperparameter | Value | Description |
| --- | --- | --- |
| Episode | 1000 | Total number of training episodes |
| Max steps | 60 | Maximum number of steps per episode |
| Number of actions | 4 | Number of different actions |
| Discount factor | 0.99 | Discount factor of future rewards |
| Learning rate | 0.00025 | Learning rate used by the optimizer |
| Initial $\epsilon$ | 1 | Initial value for exploration |
| Final $\epsilon$ | 0.1 | Minimum value for exploration |
| Target update | 4 | Step frequency to update the target network |

*4.4. DDPG*

The second algorithm we want to study to optimize ATFM delays is DDPG. We follow the approach presented in [30], which adapts the ideas underlying the success of DQN to continuous actions. DDPG is an actor–critic method, where a parameterized actor function $\mu(s)$ specifies the current policy by mapping states to actions while the critic $Q(s,a)$ learns how good is the action. Similarly to our previous approach, this implementation of DDPG directly learns from raw pixel information, using a replay buffer, and throughout the use of target networks (one for the actor and one for the critic).

The chosen NN for the actor takes as input $150 \times 100 \times 3$ images. The first layer convolves 32 ($8 \times 8$) filters with stride 4 and uses a ReLU activation function. The second layer is a batch normalization layer. The third layer convolves 64 ($4 \times 4$) filters with stride 2 and uses a ReLU activation function. The fourth layer is a batch normalization layer. The final hidden layers are a fully-connected with 256 rectifier units and a droupout layer with a rate of 0.5. The output layer is a fully-connected linear layer with a single output unit.

The chosen NN for the critic takes as input $150 \times 100 \times 3$ images and the action predicted by the actor. The first layer convolves 32 ($8 \times 8$) filters with stride 4 and uses a rectified linear unit (ReLU) activation function. The second layer is a batch normalization layer. The third layer convolves 64 ($4 \times 4$) filters with stride 2 with a rectified linear unit (ReLU) activation function. The fourth layer is a batch normalization layer. The fifth layer is a fully-connected with 256 rectifier units. The sixth layer is fully-connected with 128 rectifier units and takes as input the concatenation of the output from the fifth layer and the action from the actor. The output layer is a fully-connected linear layer with a single output unit.

A major challenge of learning in continuous action spaces is exploration. An advantage of off-policy algorithms, such as DDPG, is that we can treat the exploration problem independently from the learning algorithm. We constructed an exploration policy $\mu'$ by adding noise sampled from a noise process $\mathcal{J}$ to our actor policy:

$$\mu'(s_t) = \mu(s_t) + \mathcal{J} \tag{17}$$

where $\mu'(s_t)$ is the noised policy, $\mu(s_t)$ is the current policy, and $\mathcal{J}$ is the action noise.

In the first published article based on DDPG and raw pixel images, the authors used the stochastic Ornstein–Uhlenbeck process [42] to generate random values temporally correlated as action noise. However, in the literature, we can also find implementations using exploratory noise from a normal distribution. Although these exploration approaches are proven to work, recent studies claim that parameter noise frequently boosts performance [43]. Parameter noise adds adaptive noise to the parameters of the NN policy (actor). It injects randomness directly on the weight of the NN, altering the type of actions the agent makes depending on what the agent currently senses. Different layers of the NN have different sensitivities to perturbation, which is why we add parameter noise to the last fully connected layers. In this paper, we will analyze the performance of the models using all three types of noise.

Finally, batches of 64 random samples are used from a replay buffer of size 25,000 to train the networks. Online actor and critic networks are trained by computing the target value and respective loss. The loss is the minimum squared error of the predicted and target values. The optimizer used is Adam. The actor and critic target networks are updated using soft target updates instead of directly copying the weights. Table 2 shows the values of the remaining hyperparameters.

**Table 2.** Hyperparameters for the DDPG algorithm.

| Hyperparameter | Value | Description |
|---|---|---|
| Episode | 1000 | Total number of training episodes |
| Max steps | 60 | Maximum number of steps per episode |
| Discount factor | 0.99 | Discount factor of future rewards |
| Learning rate actor | 0.001 | Learning rate used by the optimizer |
| Learning rate critic | 0.002 | Learning rate used by the optimizer |
| Initial $\epsilon$ | 1 | Initial value for exploration |
| Final $\epsilon$ | 0.1 | Minimum value for exploration |
| Target update | 4 | Step frequency to update the target network |

## 5. Results and Discussion

This section presents the results obtained for both DQN and DDPG implementations, learning from raw pixel images to assign ATFM delay, with the dataset described in Section 4.1.

### 5.1. Performance Evaluation

The KPI is defined to evaluate the quality of the ATFM delay policy:

- The sum of the rewards received by all the agents.
- The sum of THE ATFM delay imposed by the agents.
- The total number of delayed flights.
- The sum of times the agents delayed a flight.
- The mean OC of the sector along the episode.

These KPI's are all relevant when evaluating the ATFM plan on a MARL system. One of the most widely used indicators to evaluate the performance of the agents is the sum of rewards earned at the end of each episode. The total delay imposed by the flights is also key because it is one of the indicators to minimize. The total number of delayed flights and the number of times the agents applied a delay (number of actions) can be considered KPIs showing how those delays are distributed among aircraft and the number of micro-adjustments agents make. The OC is key because it dictates situations with severe demand–capacity imbalances.

To compare the performance between the different implementations, Figure 3 shows the trend of the different KPIs using a moving window of fifty episodes. Those values have been obtained in all the cases, periodically testing the policy without exploratory noise.

The results demonstrate the potential of using RL algorithms based on images to solve DCB problems. As expected, the total reward per episode increases with the number of episodes, meaning that the agents are able to improve their policy by gathering experience from the environment. For the last 250 episodes, where we can assume convergence of the reward, DQN reported a reward of around −3000 while DDPG is around 1500. Note that the reward will always be smaller than 0 because the scenarios the agents will see always have DCB issues; thus, ATFM delay is mandatory. From the point of view of maximizing the cumulative reward, DDPG exhibits better performance than DQN.

The total ATFM delay shows a downward trend, denoting that the agents can infer which flights are more efficient to delay. DQN is the algorithm with the largest delay in the last episodes. DDPG with exploratory noise from a normal distribution reported the lowest delay while DDPG with Ornstein–Uhlenbeck and parameter noise reported an intermediate amount of delay. The main reason behind this difference in performance could come from the native characteristics of the algorithms. DQN uses discrete actions, which constrains the possible delay values, while DDPG uses continuous actions providing much more flexibility.

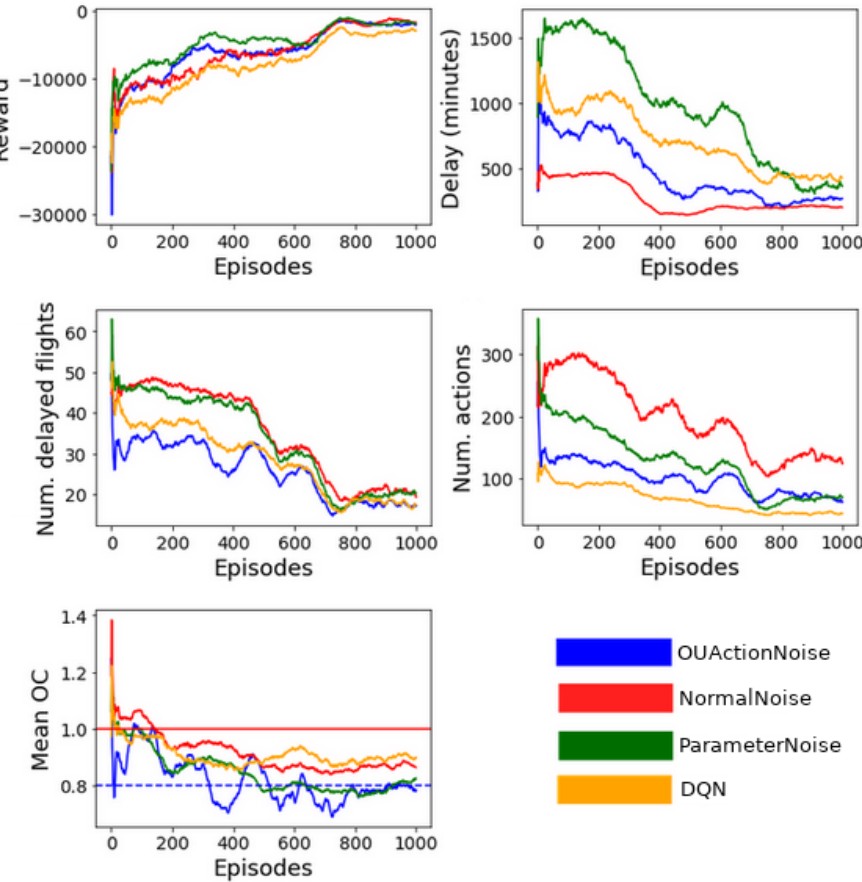

**Figure 3.** Trends KPIs used to evaluate the performances of the RL systems.

The number of delayed flights also decreases with similar behaviors between all the configurations, with an average value of around 20 delayed flights in the last 250 episodes. Although DQN and DDPG with Ornstein–Uhlenbeck seem to report slightly better performances, the improvements are minor.

Results related to the number of actions applied by the agents show that DQN is the algorithm with fewer micro-adjustments. DDPG with Ornstein–Uhlenbeck and parameter noise reported an intermediate similar number of actions. DDPG with noise from a normal distribution reported the highest value. This KPI is not directly linked to the goal of solving demand–capacity imbalances, but it is a good indicator of how many micro-adjustments are required to smooth the expected demand.

Related to the mean congestion of the sector, after 600 episodes, the sector's mean OC seems to stabilize. DDPG with Ornstein–Uhlenbeck and parameter noise reports on average an 80% usage of the airspace sector capacity, while DDPG with normal noise and DQN exhibit around 90% usage of the capacity. As a reference, in the European ATM network, the desired occupancy value is around 80% of sector capacity, providing space to absorb unexpected events and ensuring that ATCOs are not overloaded [3].

Looking at the results of the different KPIs, it is not strongly clear which approach reports the best overall performance. While DDPG with normal noise excels at reducing the overall delay, DQN or DDPG with Ornstein–Uhlenbeck achieve a greater reduction in the number of affected flights, and DDPG with Ornstein–Uhlenbeck or DDPG with parameter noise further optimize the use of sector capacity. To better analyze the behavior of the algorithms from the DCB point of view, i.e., focusing on capacity usage to smooth the expected demand, Figure 4 shows the mean OC per episode for the DDPG implementations. The image shows the collected values per episode and their trend.

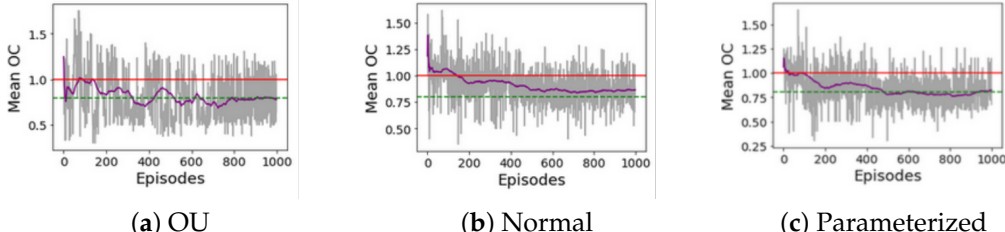

(**a**) OU        (**b**) Normal        (**c**) Parameterized

**Figure 4.** Mean occupancy count per episode for the DDPG implementations. (**a**) Ornstein–Uhlenbeck noise, (**b**) normal distribution noise, and (**c**) parameter noise. The purple line shows the trend, the red line represents the sustained capacity, and the green line represents 80% of the sector's capacity.

For the last 250 episodes where the mean OC converges, results from the DDPG with Ornstein–Uhlenbeck show the worst performance where many episodes reported a mean OC larger than the sustained capacity. DDPG with noise from a normal distribution reports slightly better results with fewer episodes with a demand greater than the sustained capacity on average. DDPG with parameter noise reports the best result with the smallest number of episodes with a mean demand larger than the sustained capacity.

Note that even though the algorithms do not keep the mean OC under the sustained capacity for all the episodes, for the last episodes where we can assume convergence in the performance, the mean demand does not exceed the peak capacity. Furthermore, focusing on the parameter noise implementation, it can be seen that the frequency and the number of consecutive episodes where the demand exceeds the sustained capacity are much smaller than in the other implementations.

Finally, a direct comparison between the results presented in this article and the actual ATFM delay is not feasible since the latter is the result of considering a broader environment. For example, let us imagine that a flight crosses two different regulated sectors. Even though the CASA algorithm could impose two different delays, the hypothetical flight would be affected only by the largest one. To directly compare the ATFM delay between the two approaches, a RL model for the two airspace sectors would be needed. Extending the proposed system to a broader region that considers the interaction between neighboring sectors is a relevant point to be studied in future works.

*5.2. Case Study*

This section presents the outcome of the framework for two specific regulations subtracted from the training dataset. The selected regulations are YBOLN07 from 7 September 2019, and YBOLN18A 18 August 2019. For each of the previous regulations, the RL system based on DDPG and parameter noise is used to collect which flights should be delayed and the amount of delay. Then, using this information, the original expected pre-tactical traffic is visualized using the following color schema:

- Red: System-suggested flights for regulation.
- Green: Non-regulated flights outside the sector in the corresponding timestamp.
- Blue: Non-regulated flights inside the sector in the corresponding timestamp.

Figure 5 shows the results for regulation YBOLN07, which started at 8:00 A.M. and finished at 10:30 A.M. As a high-level indicator, the 141 flights crossing the sector linked to regulation YBOLN07 had a total delay of 556 min (delay from YBOLN07 or any other active regulation); thus, an average delay of 3.94 min per flight. On the other hand, our RL system suggests regulating 41 (from the 141 flights crossing the traffic volume) with an average delay per flight equal to 3.71 min per flight and a maximum individual delay equal to 21 min. Note that the comparison of minutes of delay per flight considers all the regulated traffic crossing the sector independently of the regulation.

Looking at the images, the selected sector (EDYYBOLN) has two traffic flows, one from the top-left to bottom-right and another from the bottom-left to top-right. Both traffic flows are similarly regulated, indicating that the delay is spread between fights, and the

system does not have a preference. However, the RL policy sometimes decides to delay flights that do not completely cross the sector, which seems to be not ideal (see Figure 5, timestamp 8:57, red fight at the bottom-right).

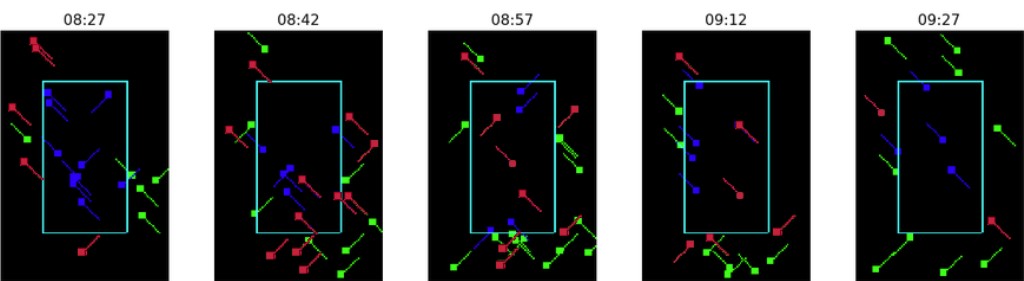

**Figure 5.** Representation of the RL system outcome for the regulation *YBOLN07*.

Note that only five images at different timestamps per regulation are shown because of space constraints. Furthermore, there are no regulated flights in the first timestamp due to the problem constraint that must guarantee each episode starts without demand–capacity imbalance (see Section 3.5).

Figure 6 shows the results for regulation YBOLN18A, which started at 2:00 P.M. and finished at 4:45 P.M. In this case, the flights crossing the sector when the regulation was active received an ATFM delay of 3.39 min per flight, while the RL framework regulated 48 (from the 159 flights crossing the traffic volume) with an average delay per flight equal to 3.35 min per flight and a maximum individual delay equal to 18 min Notice that, despite the images being more crowded than in the previous case study, the average delay per flight is slightly smaller; 3.94 versus 3.39 for the actual ATFM delay and 3.71 versus 3.35 using the RL system. This is also the case for the peak delay imposed on individual flights.

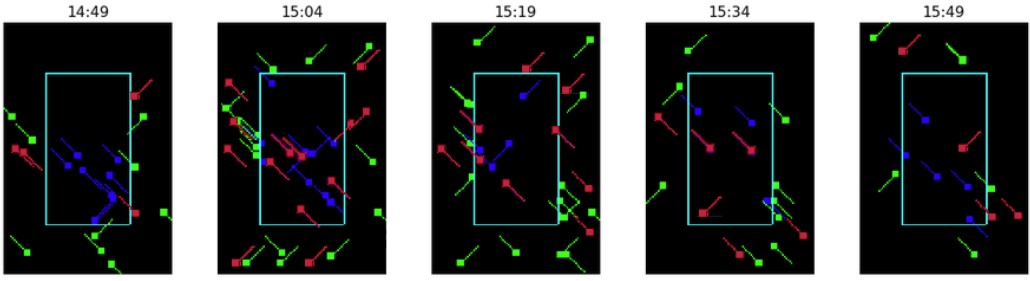

**Figure 6.** Representation of the RL system outcome for the regulation YBOLN18A.

The results obtained in these two case studies show the potential of the proposed new framework. The RL system is able to solve already detected DCB problems using images with a behavior that could be considered valid. However, a deeper analysis is required to obtain further conclusions.

## 6. Conclusions

This article proposes an image-based MARL solution to optimize ATFM delay in the European network. The goal is to maximize the usage of the airspace sector's capacity while minimizing the ground delay. The proposed approach compares DQN and DDPG algorithms with the experience replay buffer, target networks, and different strategies for exploration. Although the obtained results did not lead to a clear conclusion about which algorithm configuration best fits the problem, DDPG arises as a promising candidate. It exhibits a lower overall ATFM delay and a mean OC closer to optimal values, especially if parameter noise is used for exploration during training.

The results obtained as a first step towards devising MARL methods for deciding on ATFM delay policies using pixel images are promising. The proposed system can

successfully solve complex real-world DCB problems. Moreover, the work presented in this article could contribute to improving the usage of the airspace sector's capacity and reducing current delays.

Another relevant aspect to highlight from this research is that the approach based on images for DCB problems provides a scalable architecture that allows the representation of an arbitrary number of agents without changing the state variables architecture. This characteristic is especially relevant when working on the entire European ATM network, where thousands of flights are operated daily.

As part of future work, it becomes mandatory to study how to extend the system to larger regions of multiple sectors, thus considering the effect that the decisions made in one sector generate in the neighborhood. Additional work towards a more realistic scenario would also include using the real polygonal shape of the sectors, considering more KPIs, such as cost, or analyzing the possibility of modeling heterogeneous agents. Finally, the combination of this work with the one presented in [7] will allow us to create an end-to-end system to precisely identify the airspace sectors with demand–capacity imbalances and propose a possible solution to reduce such demand during the pre-tactical phase.

**Author Contributions:** Conceptualization, S.M.-P. and E.S.; methodology, S.M.-P. and E.S.; software, S.M.-P.; validation, S.M.-P. and E.S.; formal analysis, S.M.-P.; investigation, S.M.-P. and E.S.; resources, E.S. and E.P.; data curation, S.M.-P.; writing—original draft preparation, S.M.-P.; writing—review and editing, S.M.-P. and E.S.; visualization, S.M.-P. and E.S.; supervision, E.S.; project administration, E.S.; funding acquisition, E.S. and E.P. All authors have read and agreed to the published version of the manuscript.

**Funding:** This work was funded by EUROCONTROL under Ph.D. Research contract no. 18-220569-C2 and by the Ministry of Economy, Industry, and Competitiveness of Spain under grant number PID2020-116377RB-C21.

**Conflicts of Interest:** The authors declare no conflict of interest.

## Abbreviations

| | |
|---|---|
| AI | Artificial Intelligence |
| AIRAC | Aeronautical Information Regulation and Control |
| ANSP | Air Navigation Service Provider |
| ATC | Air Traffic Control |
| ATCO | Air Traffic Controller |
| ATFM | Air Traffic Flow Management |
| ATM | Air Traffic Management |
| CASA | Computer Assisted Slot Allocation |
| CFMU | Central Flow Management Unit |
| CNN | Convolutional Neural Network |
| CTOT | Calculated Take-Off Time |
| DCB | Demand–Capacity Balancing |
| DDPG | Deep Deterministic Policy Gradient |
| DPG | Deterministic Policy Gradient |
| DQN | Deep Q-learning |
| EATMN | European Air Traffic Management Network |
| ECAC | European Civil Aviation Conference |
| FMP | Flow Manager Position |
| KPI | Key Performance Indicator |
| MAA3C | Multi-Agent Asynchronous Advantage Actor–Critic |
| MARL | Multi-Agent Reinforcement Learning |
| MDP | Markov Decision Process |
| ML | Machine Learning |
| MUAC | Maastricht Upper Area Control Centre |
| NN | Neural Network |
| OC | Occupancy Count |

| ReLU | Rectified Linear Unit |
| RL | Reinforcement Learning |
| SOBT | Scheduled Off-Block Time |

## Nomenclature

| $\alpha$ | Learning rate |
| $\gamma$ | Discount factor |
| $\mathcal{N}$ | Number of agents |
| $\mu$ | DDPG ATFM delay policy |
| $\pi$ | DQN ATFM delay policy |
| $a$ | Action variable |
| $C$ | Capacity variable |
| $D$ | Delay variable |
| $G$ | Cumulative reward |
| $i$ | Agent indicator |
| $P$ | Transition function |
| $R$ | Reward variable |
| $r$ | Reward variable |
| $s$ | State variable |
| $TP$ | Counting time period |
| $V$ | Occupancy Count variable |

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
