# Peer review of "Image-Based Multi-Agent Reinforcement Learning for Demand–Capacity Balancing"

_aerospace, doi:10.3390/aerospace9100599_

Round 1

Reviewer 1 Report

The authors pose a standard prediction question in strategic air traffic management, and attempts to use reinforcement learning (RL) to perform this prediction. The innovation point in this article for me was the usage of the image-based state space/environment for the RL algorithm, and I believe this was an intriguing and effective way to compress what would normally cause state space/action set/etc. cardinalities to blow up exponentially. However, I do have some major concerns regarding (1) the context for the prediction problem (e.g., the authors state that strategic/pre-tactical predictions for ATFM initiatives/TMIs haven't been examined closely before, and do not contrast/differentiate their studies from the previous work on this precise topic), (2) incorporation of realism in terms of airline preferences, (3) framing their RL system in the context of trajectory-based operations, and (4) comparing solutions given by their RL system to more "traditional" approaches such as CASA.

I believe that this work is innovative and with the appropriate changes/revisions could be publishable. I would recommend a major revision. See below for more detailed comments:

(1.) Delay prediction problems where traffic management initiatives/air traffic flow management regulations are taken into consideration could (and should) be counted as research that predicts regulations during the strategic planning phase. Examples of such works include “Predicting the likelihood of airspace user rerouting to mitigate air traffic flow management delay” (Dalmau 2022) and “A Machine Learning Approach to Predict the Evolution of Air Traffic Flow Management Delay” (Dalmau et al. 2021) -- I would disagree with the authors' characterization that there hasn't been work done on the detection of such management initiatives in the pre-tactical phase.

(2.) In reality, different airlines would have different responses to assigned ground delays. For example, some may perform swaps internal to their own fleet, whereas other airlines may decide to proactively cancel to mitigate delay propagation across their schedule, etc. Could the assumption of homogeneous agents be relaxed at all? Currently, it's not certain how fleet operator preferences could be factored into account in this setup.

(3.) It's unclear how the authors propose connecting their image-based RL system to functionalities like TBO. Wouldn't an implementation reminiscent of TBO require entire traces of a flight's position to be tracked for a period of time? I think there could be a case to be made here in terms of TBO, but right now it is very unclear as to how that can happen.

(4.) Could the authors comment on the actual differences between what the RL system proposes versus what the CASA system proposes? And by this I don't mean aggregate delay statistics (e.g., RL system results in X delays versus Y delays for CASA, and X < Y), but for an "average" flight, how was its expected departure clearance time changed in the RL system versus the CASA system? An implementation of a more sophisticated CASA system (it could be something as futuristic as the RL system, or something that's a bit more of a mid-term solution such as augmented ration-by-schedule) will probably only have airline buy-in and adoption if it's shown to not produce large perturbations for individual flights.

Reviewer 2 Report

On page 6 (line 264) authors assume that “…sector’s capacity is known, and squares can be used to approximate its shape.”

Authors should explain what prompted them to simplify the shape (maybe it was a requirement of the algorithms they used to determine whether a flight is inside or outside a certain sector).

Information on flight data is to be found in various parts of the publication.

It might be worth adding a dedicated section in “4.1 Dataset” (beginning on p. 9, line 371) merging all corresponding information authors use in their analysis. It should also be explained where flight data comes from.

On page 14 (lines 566-567) authors “…mention that a direct comparison between the results presented in this article and the actual ATFM delay is not feasible”. In their subsequent case study on page 16 (lines 608-609) they evaluate their findings as follows: “The results obtained in these two case studies show that the proposed RL system for DCB problems is able to report comparable results to the actual CASA algorithm.”

In view of what was previously said, this statement could be formulated more extensively and supported with (additional) facts.

Typos / Misc.

p. 2 (line 48) “face” to be replaced by “phase”

p. 9 (line 342) “Problem Constraints” instead of “Problem Constrain”?

p. 9 (line 379) Is "EDDYBOLN” meant to be “EDYYBOLN”?

p. 15 (line 578) “from August 18th 2029” supposed to be 2019?

p. 18 (lines 704-705) hyperlink does not work due to line break

Round 2

Reviewer 1 Report

The authors have addressed all of my comments to my satisfaction -- I stand by my original assessment that this is an innovative approach, via combining image-based RL and ATFM/DCB. I am happy to recommend publication.